

# Development of a high-density sub-species-specific targeted SNP assay for Rocky Mountain bighorn sheep (*Ovis canadensis canadensis*)

Samuel Deakin[1] and David W. Coltman[1,2]

[1] Department of Biological Sciences, University of Alberta, Edmonton, Alberta, Canada
[2] Department of Biology, University of Western Ontario, London, Ontario, Canada

## ABSTRACT

Due to their abundance and relative ease of genotyping, single nucleotide polymorphisms (SNPs) are a commonly used molecular marker for contemporary population genetic and genomic studies. A high-density and cost-effective way to type SNP loci is Allegro targeted genotyping (ATG), which is a form of targeted genotyping by sequencing developed and offered by Tecan genomics. One major drawback of this technology is the need for a reference genome and information on SNP loci when designing a SNP assay. However, for some non-model species genomic information from other closely related species can be used. Here we describe our process of developing an ATG assay to target 50,000 SNPs in Rocky Mountain bighorn sheep, using a reference genome from domestic sheep and SNP resources from prior bighorn sheep studies. We successfully developed a high accuracy, high-density, and relatively low-cost SNP assay for genotyping Rocky Mountain bighorn sheep that genotyped ~45,000 SNP loci. These loci were relatively evenly distributed throughout the genome. Furthermore, the assay produced genotypes at tens of thousands of SNP loci when tested on other mountain sheep species and subspecies.

# INTRODUCTION

Molecular markers are essential tools in the field of molecular ecology. Due to their abundance and relative ease of genotyping, single nucleotide polymorphisms (SNPs) are a common marker-type of choice for contemporary population genetic and genomic studies (*Grover & Sharma, 2016*). Despite typically being biallelic and therefore less informative than multiallelic markers such as microsatellites (*Aitken et al., 2004*), their relative ease of discovery in both model and non-model organisms (*Baird et al., 2008*; *Peterson et al., 2012*; *Narum et al., 2013*), abundance (*Aitken et al., 2004*), and ability to be sequenced on extremely high throughput next-generation sequencing (NGS) platforms (*Metzker, 2010*) make SNPs an ideal marker for high-density genotyping of large numbers of individuals.

There are multiple approaches to typing large numbers of SNPs in non-model organisms. For species without a reference genome sequence, *de novo* discovery by genotyping by sequencing (GBS) (*Narum et al., 2013*) approaches such as restriction-site

Corresponding author
Samuel Deakin,
samuel.deakin@ucalgary.ca

associated DNA sequencing (RAD-seq) (*Baird et al., 2008*) and double digest restriction-site associated DNA sequencing (ddRAD-seq) (*Peterson et al., 2012*) can be used. However, these methods are not targeted to specific genomic regions. An alternative to RAD-seq and ddRAD-seq is targeted-GBS, which allows user-specified regions of the genome to be sequenced (*Kozarewa et al., 2015*; *Meek & Larson, 2019*; *Scaglione et al., 2019*). Allegro targeted genotyping (ATG) is a recently developed, high-density, cost-effective form of targeted GBS offered by Tecan genomics (Redwood City, CA, USA) that utilizes Single Primer Enrichment Technology (SPET) (*Barchi et al., 2019*; *Scaglione et al., 2019*). This form of SPET uses a simplified single primer design and functions by sequencing flanking regions around a probe to sequence target regions of interest (*Barchi et al., 2019*). Despite being a relatively new technology, it has proven applications in humans (*Nairismägi et al., 2016*; *Saber et al., 2017*; *Scolnick et al., 2015*), non-human mammals (*Andrews et al., 2021*; *Gavriliuc et al., 2022*), plants (*Barchi et al., 2019*; *Gramazio et al., 2020*; *Scaglione et al., 2019*), arthropods (*Chang et al., 2020*; *Vu et al., 2023*), and bacteria (*Benjamino et al., 2021*; *Homeier-Bachmann et al., 2022*).

One major drawback of all targeted-GBS technologies is the genomic information required to design an assay. Unlike RAD based methods, targeted-GBS requires prior knowledge of both polymorphic loci or genomic regions of interest and a reference genome applicable to the target species, and therefore may not be suitable for all species (*Kozarewa et al., 2015*). However, for some non-model species genomic information from other closely related species can be used. For example, genomic resources developed for the domestic cat and sheep have been applied to their wild counterparts (*Li et al., 2019*; *Santos et al., 2021*; *Sim & Coltman, 2019*). Thus, for wild species for which a genome of a closely related species and SNP or genomic regions of interest are known for the target species or closely related species, targeted-GBS is a viable high-density SNP genotyping technology.

The Rocky Mountain bighorn sheep (*Ovis canadensis canadensis*) may be a good candidate for the application of Allegro targeted-GBS for two reasons. Firstly, a reference genome exists for the closely related domestic sheep (*Ovis aries*) which diverged from bighorn sheep ~3 mya (*Bunch et al., 2006*), and previous cross-species applications of the resources developed for domestic sheep to bighorn sheep genomic studies have been successful. Secondly, multiple studies have previously identified abundant SNP loci in Rocky Mountain bighorn sheep (*Kardos et al., 2015*; *Miller, Festa-Bianchet & Coltman, 2018*; *Miller et al., 2015*, p. 201; *Miller, Hogg & Coltman, 2013*; *Miller et al., 2012*). Thus, all the resources required to develop a targeted-GBS assay are available. Additionally, Tecan's ATG requires modest quantities of genomic DNA which make it well suited for field studies relying on non-invasively collected sample types such as faeces (*Albaugh et al., 1992*; *Gavriliuc et al., 2022*).

Here, we describe the development of a high-density SNP assay to be used with the ATG technology for Rocky Mountain bighorn sheep in western Canada. Our aim was to target 50,000 SNP loci in one assay which can be used for population genetic, genomic, and quantitative genetic studies. We also test the efficiency and accuracy of the assay by sequencing multiple individuals on a trial and full version of our assay. We include template DNA from differing biological materials to test how DNA source affects the

accuracy of the assay. Finally, although not a focus of this study, we also examine the assay performance on two other subspecies of bighorn sheep; desert (*Ovis canadensis nelsoni*) and Sierra Nevada (*Ovis canadensis sierrae*) bighorn sheep which diverged from Rocky Mountain bighorn sheep ~94 and ~315 kya, respectively (*Buchalski et al., 2016*), and both subspecies of thinhorn sheep; Dall (*Ovis dalli dalli*) and Stone (*Ovis dalli stonei*) sheep which diverged from bighorn sheep ~1 mya (*Rezaei et al., 2010*).

## METHODS

### Assay development

Variant site data were sourced from five studies; an implementation of a domestic sheep SNP-chip (*Miller et al., 2012*), RAD-seq performed on individuals from Ram Mountain, Alberta and the National Bison Range, Montana (*Miller, Hogg & Coltman, 2013*) (Dryad: doi: 10.5061/dryad.4qk81), whole genome sequencing performed on one individual from Ram Mountain, Alberta (*Miller et al., 2015*) (NCBI SRA: SRP052039), pooled whole genome sequencing of 58 individuals from Montana and Wyoming (*Kardos et al., 2015*) (Dryad: doi: 10.5061/dryad.3f2t2), and the application of a high density domestic sheep SNP-chip (*Miller, Festa-Bianchet & Coltman, 2018*) (Dryad: doi: 10.5061/dryad.c0p090f). We took the variant call data files (VCFs) from four of these studies; *Kardos et al. (2015)*, *Miller et al. (2012)*, *(2015)*, and *Miller, Festa-Bianchet & Coltman (2018)*, as these studies were already mapped to the *Ovis aries 3.1* (OAR3.1) reference genome (*Jiang et al., 2014*) (NCBI refseq ID: GCF_000298735.1). The *Miller, Hogg & Coltman (2013)* study was mapped to an older domestic sheep reference genome, therefore we aligned its raw reads to the OAR3.1 reference genome, and then called variants from this alignment to produce a variant call file in a format consistent with that of the other studies. To expedite and reduce computational power required for alignment of these reads, prior to alignment we created a reduced version the OAR3.1 genome. This reduced genome only included sequence data for a 400 bp region, 200 bp upstream to 200 bp downstream, of each of the 50,000 target loci. We produced indexes from this reduced genome using the Bowtie2 v2.3.5.1 (*Langmead & Salzberg, 2012*) build function and the SAMtools v1.10 (*Li et al., 2009*) command "faidx". Then both forward and reverse reads were aligned as pairs to the indexed reference genome using the "—sensitive" pre-set options in Bowtie2. The SAMtools command "view" was then run, with the reference index specified, to convert the paired read alignments from un-indexed .sam files to indexed .bam files. Finally, the SAMtools command "sort" was used to sort the .bam files.

We subsequently filtered the variant site data generated by aligning the *Miller, Hogg & Coltman (2013)* reads and variant data from the other four studies through a series of steps to reduce the number of variants and obtain optimal loci for our assay. To filter the variant site data we used VCFtools v0.1.15 (*Danecek et al., 2011*) and BCFtools v1.10.2 (*Li et al., 2009*). The filtering process is illustrated in Fig. 1. We used the same filtering processes for variants sourced from all five studies with one exception. Since the *Kardos et al. (2015)* sampling locations were distant from western Canadian populations, to maximize targeting loci polymorphic across populations we filtered out loci with a minor allele

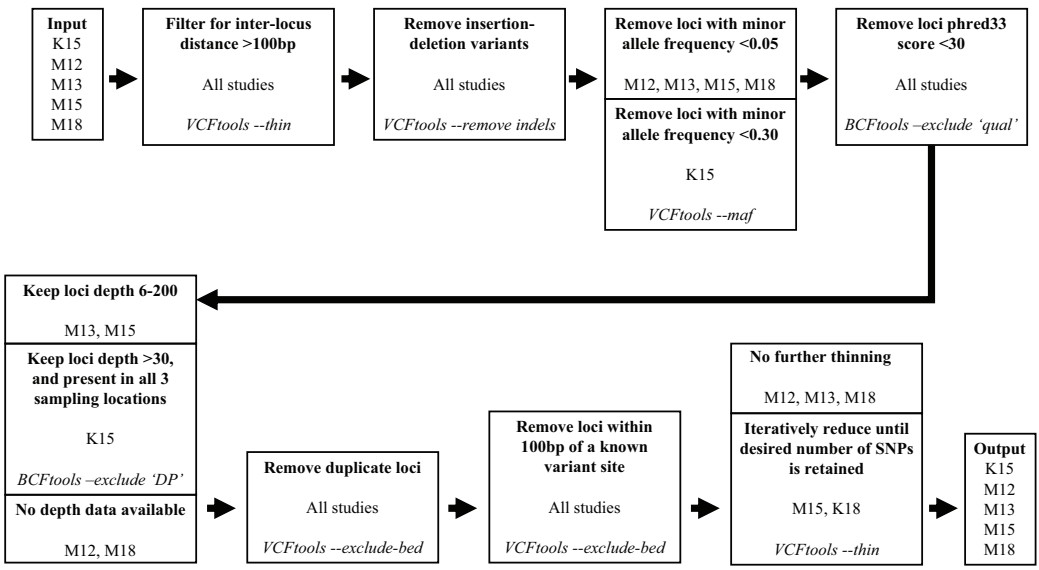

**Figure 1  Filtering steps applied to variant data sourced from the five studies.** The studies *Miller et al. (2012)*, *Miller, Hogg & Coltman (2013)*, *Miller et al. (2015)*, *Kardos et al. (2015)*, and *Miller, Festa-Bianchet & Coltman (2018)* are represented by M12, M13, M15, K15, and M18, respectively. Each step describes a process applied to data from the previous step, process is indicated in bold, source studies of variant data in plain text, and software and command used in italics.

frequency <0.3 and only retained loci polymorphic in all three (*Kardos et al. 2015*) sampling locations.

Our target was to submit 100,000 candidate SNPs to Tecan for probe design, as the probe design process had ~50% success rate in designing paired probes for target loci. To reach this target we retained all the *Miller et al. (2012)*, *Miller, Hogg & Coltman (2013)*, and *Miller, Festa-Bianchet & Coltman (2018)* SNPs that passed filtering, and subsampled SNPs from the *Kardos et al. (2015)* and *Miller et al. (2015)* data which passed filtering. Subsampling of the *Kardos et al. (2015)* and *Miller et al. (2015)* SNPs was done by using the VCFtools function "—thin" at increasingly larger increments until we reduced the candidate SNP list to ~100,000 (Fig. 1). We submitted the target SNP list to Tecan as a regions file (bed format), targeting a 5 base pair (bp) region around our target SNPs (2 bp upstream, 2 bp downstream), from which they designed a genotyping assay for 50,000 loci across the Rocky Mountain bighorn sheep genome based upon the OAR3.1 domestic sheep genome. We visualised SNP distribution in a rainfall plot generated using the R (*R Core Team, 2013*) package karyoploteR (*Gel & Serra, 2017*). We also plotted chromosomal coverage and inter-SNP distances using ggplot2 v3.3.5 (*Wickham, 2011*).

Prior to confirming the 50,000 loci assay (the "50k assay") design, we developed a 10,000 loci trial assay (the "10k assay") to test the ATG technology. We genotyped a common set of individuals with both assays to assess performance. This comprised DNA samples from 56 individuals from five species and subspecies of mountain sheep. All Rocky Mountain bighorn sheep samples were typed in duplicate to test the accuracy of the assays, yielding 96 samples genotyped (Table 1). To test the performance of the assay on differing

**Table 1 Sample information for the 56 individuals from multiple species and subspecies of mountain sheep typed on the 10,000 SNP and 50,000 SNP validation runs.**

| Species/sub-species | Sample type | Individuals genotyped | Number of replicates | Number of samples genotyped |
|---|---|---|---|---|
| Rocky Mountain bighorn sheep | Blood, Faeces, Tissue | 40 | 2 | 80 |
| Sierra Nevada bighorn sheep | Tissue | 4 | 1 | 4 |
| Desert bighorn sheep | Tissue | 4 | 1 | 4 |
| Dall sheep | Tissue | 4 | 1 | 4 |
| Stone sheep | Tissue | 4 | 1 | 4 |
| Total | | 56 | | 96 |

**Table 2 Locations and number of samples for the Rocky Mountain bighorn sheep typed in the 96-sample validation run.**

| Sampling location | Abbreviation | Sample type | Individuals genotyped | Number of replicates | Latitude (°N) | Longitude (°W) |
|---|---|---|---|---|---|---|
| Cadomin Mountain | CM | Faeces | 8 | 2 | 52.97 | 117.20 |
| Castle Yarrow | CY | Blood | 8 | 2 | 49.27 | 114.20 |
| Narraway | NW | Tissue | 8 | 2 | 54.27 | 119.91 |
| Ram Mountain | RM | Tissue | 8 | 2 | 52.36 | 115.79 |
| Stornoway Mountain | ST | Faeces | 8 | 2 | 53.29 | 118.39 |

biological sample types, Rocky Mountain bighorn sheep DNA was extracted from blood, faeces, and other tissues (*i.e.*, skin and muscle) (Table 2, Fig. 2).

We extracted DNA from differing biological material using different protocols for each material type. Biological materials were collected from wild animals under protocols approved by the University of Alberta (certificate No. 610901), University of Calgary (protocol No. BI11R-14), and Université de Sherbrooke Animal Care Committee (Protocol MFB01). Tissue samples were digested and extracted using the DNeasy Blood and Tissue kit (Qiagen N.V., Venlo, the Netherlands), following the manufacturer's recommended procedure. Blood was treated with ammonium–chloride–potassium (ACK) (*Brown, Hu & Athanasiou, 2016*) prior to undergoing the same procedure as other tissue samples. For faecal samples, three pellets from each sample were soaked in 1X phosphate-buffered saline (PBS) for 20 min then swabbed with a cotton-tip applicator. The applicator was washed with Aquastool (MoBiTec GmbH, Goettingen, Germany) then DNA extracted following the Aquastool protocol. Each sample was quantified and normalized to a concentration of 8 ng/uL, resulting in a total of 80 ng of DNA processed per sample during library preparation. For the library preparation we used two Allegro Targeted Genotyping V1 kits from Tecan Genomics. The first kit targeted the 10k assay, which was a subset of SNP loci from the full 50k assay, and the second kit targeted the 50k assay. We followed the manufacturer's protocol provided by Tecan with the fragmentation digest time increased from 15 to 22.5 min. Following library preparation, we sequenced a total of 96 samples: 80 Rocky Mountain bighorn sheep (40 individuals in duplicate), four desert bighorn sheep, four Dall sheep, four Sierra Nevada bighorn sheep, and four Stone sheep using both kits.
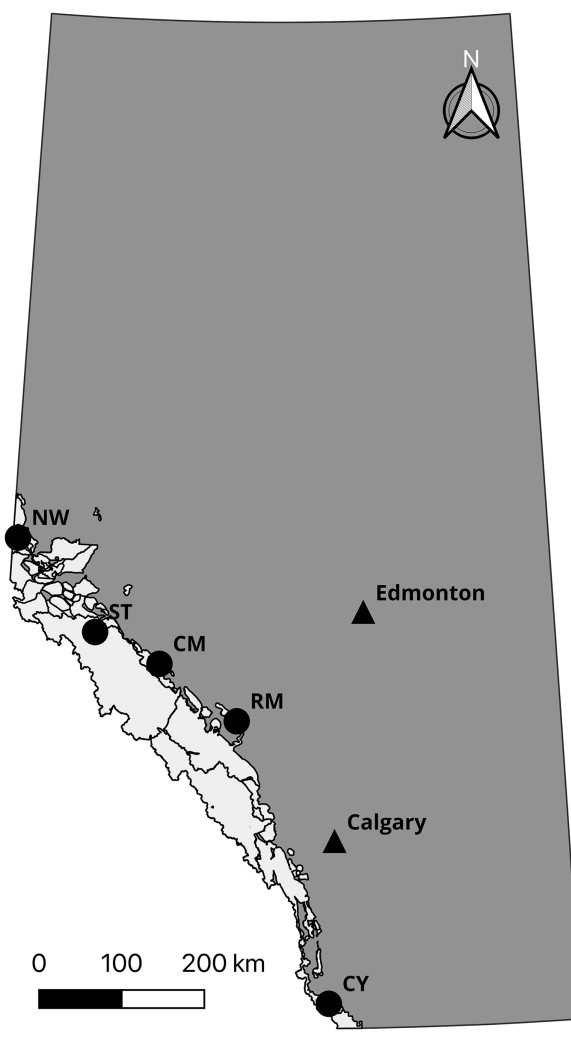

**Figure 2 Sampling locations of Rocky Mountain bighorn sheep genotyped on the 10 and 50k assays.** Rocky Mountain bighorn sheep ranges in Alberta is shown in light grey. Sampling location abbreviations: Cadomin Mountain (CM), Castle Yarrow (CY), Narraway (NW), Ram Mountain (RM), Stornoway (ST). Contains information licensed under the Open Government Licence—Canada and Alberta.

The 10k assay was sequenced on an Illumina (San Diego, CA, USA) NextSeq using a NextSeq V2.5 mid-output 300 cycle kit which provided 120,000,000 two-directional 150 bp reads. Thus, each sample had an average coverage of 125X (120,000,000 reads ÷ 10k loci ÷ 96 samples = 125X coverage). The 50k assay samples were sequenced as part of a 384-individual library on a NextSeq V2.5 high-output 300 cycle kit providing 400,000,000 two-directional 150 bp reads. Thus, each sample had an average coverage of 20.83X (400,000,000 reads ÷ 50k loci ÷ 384 samples = 20.83X coverage).

The raw reads generated by both sequencing runs were analysed using FastQC v0.11.9 (*Andrews, 2010*). Raw reads were trimmed using TrimGalore v0.6.5 (*Martin, 2011*) with the commands "—paired" and "—phred33". Additionally, the command "—adapter2"

followed by adaptor sequence removed adaptors from the R2 read. The commands "—three_prime_clip_R1/R2 5" and "—quality 30" were used to trim five bp from the three-prime end of both the R1 and R2 reads, and trim anything with a quality less than 30. We divided our reads by sequencing run (for the 10 or 50k assay) and by subspecies prior to calling variant sites.

To align the reads to the OAR3.1 genome and call variant sites, we used Bowtie2, SAMtools, and BCFtools following the same process and parameters as previously described for realigning the *Miller, Hogg & Coltman (2013)* reads. Subsequently, we filtered the 10 VCF files individually. We removed indels using VCFtools, then removed loci with a quality score lower than 30 and a depth of less than six using BCFtools. We excluded loci with more than 30% missing data and loci with a minor allele frequency of less than 0.05 in Rocky Mountain bighorn sheep and 0.15 in all other species using VCFtools. Samples with more than 25% missing data were removed, and the resulting dataset was re-filtered based on the earlier minor allele frequency requirements to ensure all retained SNPs still had suitable minor allele frequencies using VCFtools. Finally, we filtered to retain loci with no more than 10% missing data using VCFtools.

## Assay validation

To evaluate the efficiency of the two assays, we examined reads from the 56 genotyped individuals, 96 samples in total including duplicates, sequenced on the 10 and 50k assays. First, we examined the overall efficiency and on-target efficiency of each assay. We defined the overall efficiency as the total number of SNPs recovered relative to the number of loci targeted, which, due to the nature of the SPET technology, could be higher than 100% as novel SNPs may be discovered in the flanking regions of the targeted loci. We defined on-target efficiency as the number of targeted SNPs recovered relative to the number of loci targeted.

To evaluate accuracy, we compared genotypes between duplicate samples within and between the two assays. We used PLINK (*Purcell et al., 2007*) to calculate the identity by state (IBS) for all genotypes, calculated as a proportion of identical loci between two samples excluding missing data. Rocky Mountain bighorn sheep duplicate samples were compared within and between the two assays. Other subspecies were only compared between the two assays which shared common genotypes. To compare the accuracy on DNA extracted from differing biological materials, we separated the Rocky Mountain bighorn sheep samples by type; blood, faecal, and other tissue derived DNA. To further evaluate the accuracy of our assays we examined the concordance between genotypes generated from our 10K and 50K panels and the genotypes produced in *Miller, Hogg & Coltman (2013)* and *Miller, Festa-Bianchet & Coltman (2018)*. To do this we identified individuals genotyped in both our and *Miller, Hogg & Coltman (2013)* and/or *Miller, Festa-Bianchet & Coltman (2018)* and then calculated the identity by state between the pairs of genotypes using PLINK.

We used PLINK to calculate observed ($H_{obs}$) and expected heterozygosity ($H_{exp}$) within sampling locations of the Rocky Mountain bighorn sheep, as well as the total sample set, using only one of the duplicated samples. We then calculated Wright's inbreeding

**Table 3 Source studies of SNPs and number of from each study SNPs retained in the 50k assay.**

| Study | Total variant sites | SNPs retained in 50k assay |
|---|---|---|
| Kardos et al. (2015) | 20,117,094 | 21,121 |
| Miller et al. (2012) | 40,843 | 147 |
| Miller, Hogg & Coltman (2013) | 290,287 | 4,244 |
| Miller et al. (2015) | 19,153.582 | 23,829 |
| Miller, Festa-Bianchet & Coltman (2018) | 3,777 | 659 |

coefficient $f = (H_{exp} - H_{obs})/H_{exp}$ (*Wright, 1922*). Additionally, we calculated minor allele frequencies for all SNPs using PLINK. *Nei*'s *(1972)* standard genetic distances between Rocky Mountain bighorn sheep sampling locations were calculated in adegenet (*Jombart, 2008*; *Jombart & Ahmed, 2011*). We then used the R package vegan v2.5.7 (*Oksanen et al., 2013*) to perform simple Mantel tests on the genetic distances and geographic distances between sampling locations. Finally, to illustrate the ability of each assay to resolve population structure, we plotted the first two principal components extracted from principal component analyses (PCA) of Rocky Mountain bighorn sheep data using the R package adegenet. As samples were genotyped in duplicate, we expect individuals successfully genotyped in duplicate to overlap or be in close proximity on the PCA, depending on the error rate between duplicates.

# RESULTS

## Assay development

From the five studies we sourced a total of ~39.7 million variant sites. After filtering, we retained 100,000 SNP loci and submitted them to Tecan for assay development. From this list Tecan designed an assay which could target 50,000 loci. For more information on the numbers of loci sourced from each study and included in the 50k assay see Table 3, for full details of the assay designed by Tecan see electronic material 1. The 50,000 loci were relatively evenly distributed within and between chromosomes (Figs. 3 and 4) and had an average inter-loci distance of 51,663 bp ± 60,107 (Fig. 5) (based upon inter-loci distances from the *Jiang et al. (2014)* genome assembly).

## Post-sequencing filtering

After trimming and alignment, we identified 163,176 and 769,835 variant sites from the 10 and 50k assays, respectively, in Rocky Mountain bighorn sheep. After filtering we retained 10,215 and 45,367 SNPs in the 10 and 50k datasets, respectively, of which 8,082 and 38,499 SNPs were "on-target" in the 10 and 50k SNP datasets, respectively (Table 4). For further details of SNPs retained throughout filtering steps in the other mountain sheep species and subspecies, see Table 4.

## Assay validation

The 10 and 50k assays had overall efficiencies 102% and 91%, respectively, for Rocky Mountain bighorn sheep samples. On-target efficiency was lower at 81% and 77% for the
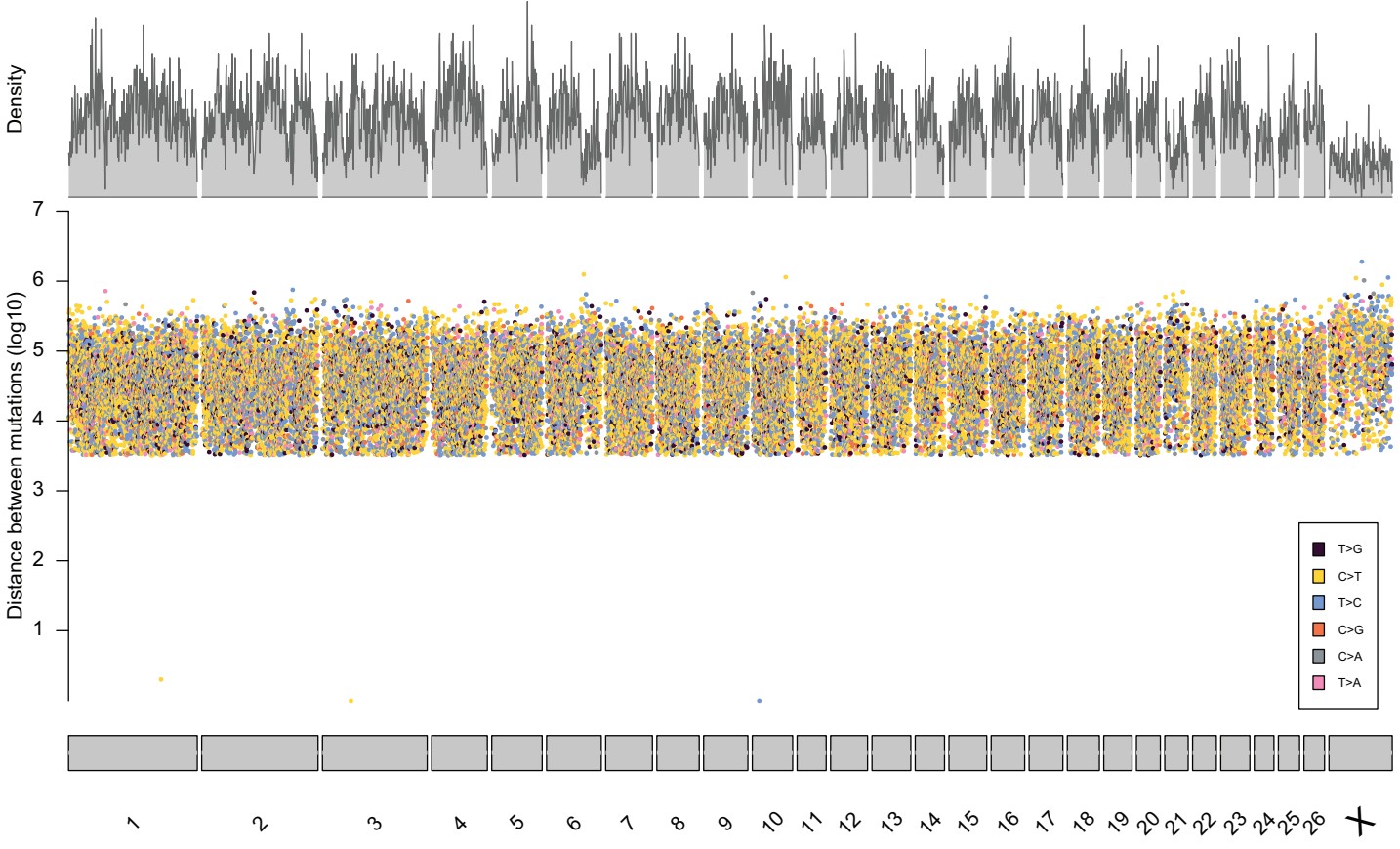

**Figure 3** **Rainfall plot characterising the position, inter-loci distance, and type of substitution for each of the 50,000 loci targeted by the 50k assay.** Density across chromosome regions shown in grey.

10 and 50k loci assays, respectively, based on Rocky Mountain bighorn sheep samples. For other subspecies, overall efficiency ranged from 32% to 43% and 17% to 42%, respectively, and on-target efficiency ranged from 14% to 29% and 12% to 31% for the 10 and 50k SNP assays, respectively.

IBS between Rocky Mountain bighorn sheep duplicate samples ranged from 96% to 97% within and between the 10 and 50k SNP assays (Table 4). The values of IBS between assays for other species and subspecies of mountain sheep ranged from 93% to 96% (Table 5). The IBS of genotypes from DNA extracted from blood and other tissue exceeded those for DNA from faeces (Table 6). The only sample type to lose sufficient numbers of genotypes during filtering to cause samples to fail quality control were faecal material, with the 10k and 50k assays losing 21% and 56% of samples, respectively (Table 6).

Rocky Mountain bighorn sheep samples had observed heterozygosity of 28% and 29% and expected heterozygosity of 33% and 33% for the 10 and 50k SNP assays, respectively. Wright's inbreeding coefficient for the pooled Rocky Mountain sheep sample set was 0.133 and 0.135 for the 10k SNP and 50k SNP assays, respectively (Table 7). There was a strong pattern of isolation-by-distance among the five sampling locations of Rocky Mountain bighorn sheep in both SNP assays (Fig. 6), with geographic distance accounting for 88%
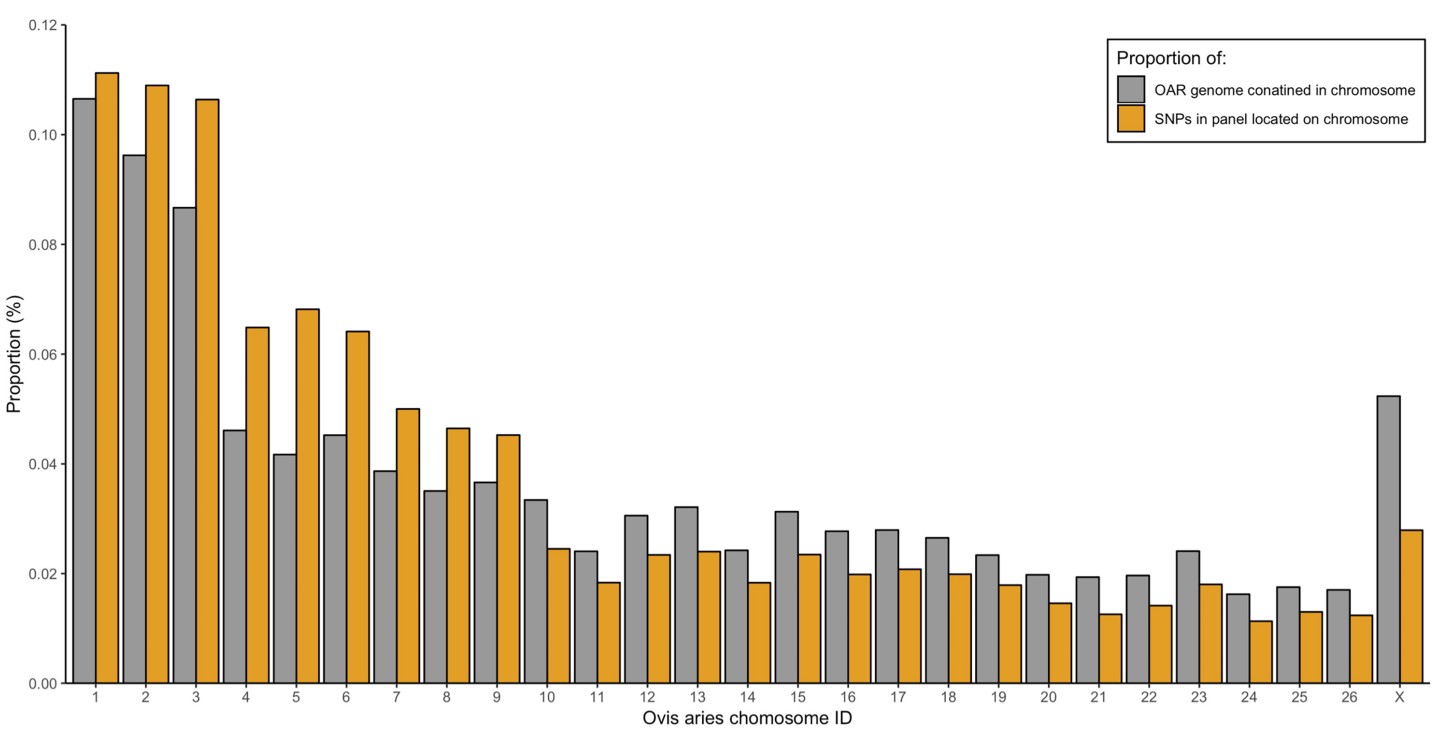

**Figure 4 Proportion of the total *Ovis aries* 3.1 genome contained within each chromosome (grey) alongside the proportion of SNPs in the assay located on each chromosome (gold).**

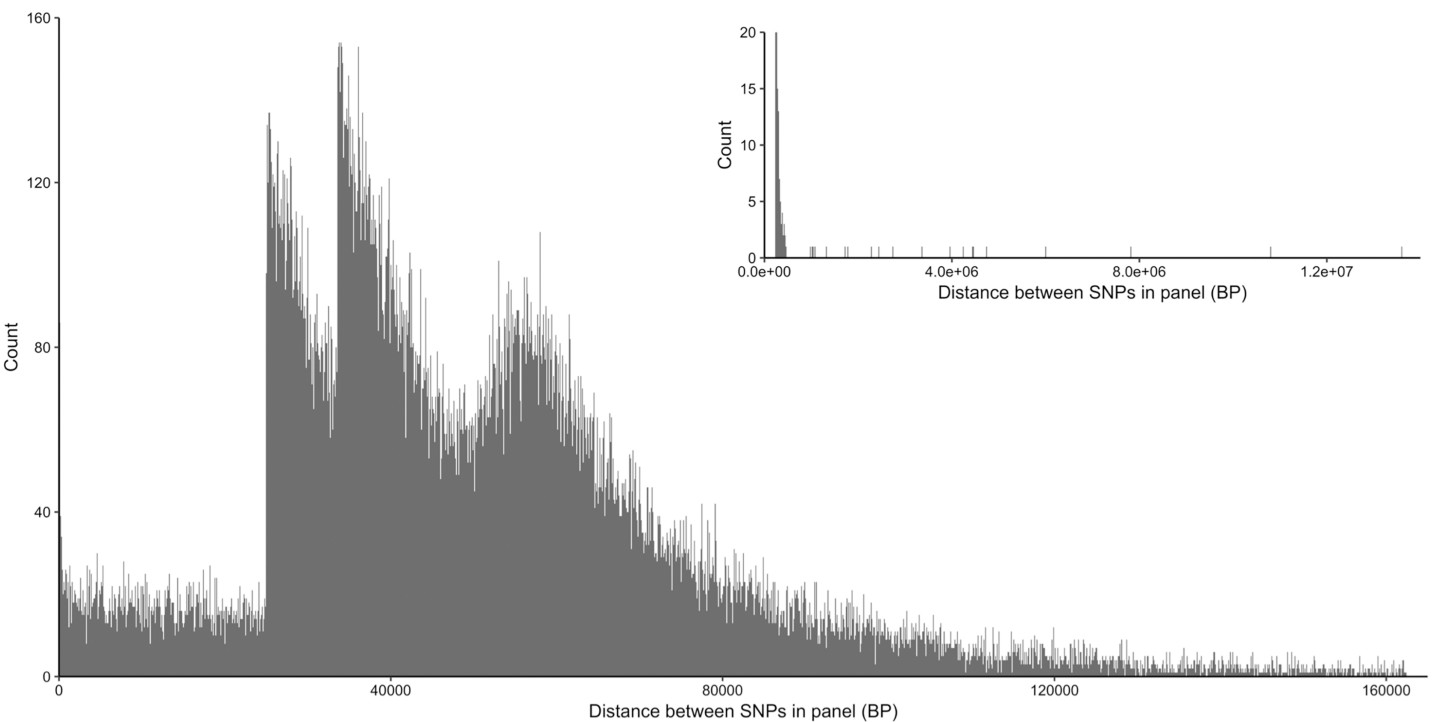

**Figure 5 Distribution of inter-loci distances in base pairs between neighboring SNP loci in the 50k assay.** Main plot shows 99% of the data excluding the longest 1% of distances, insert (top right) shows all data.

**Table 4 Variant sites retained after each stage of filtering and efficiencies for the five species/subspecies of mountain sheep genotyped on the 10,000 and 50,000 SNP assays.**

| Species/sub-species | 10,000 SNP assay | | | | | 50,000 SNP assay | | | | |
|---|---|---|---|---|---|---|---|---|---|---|
| | RM bighorn | Dall | Desert | SN bighorn | Stone | RM bighorn | Dall | Desert | SN bighorn | Stone |
| All sites | 163,176 | 67,212 | 65,849 | 64,838 | 62,102 | 769,835 | 140,008 | 131,293 | 126,914 | 128,393 |
| Remove indels | 89,879 | 12,699 | 14,503 | 13,677 | 12,999 | 449,563 | 82,379 | 81,474 | 79,508 | 85,747 |
| Quality >30 | 49,815 | 9,977 | 11,476 | 10,814 | 9,770 | 183,799 | 35,471 | 39,925 | 38,182 | 31,876 |
| Depth >6 | 49,059 | 9,778 | 11,230 | 10,560 | 9,528 | 183,476 | 27,495 | 31,515 | 29,495 | 22,923 |
| Loci missing data <30% | 45,736 | 9,682 | 11,106 | 10,445 | 9,395 | 148,161 | 26,556 | 28,889 | 27,208 | 21,915 |
| Minor allele frequency >5% | 11,502 | 3,431 | 4,440 | 3,769 | 3,338 | 48,791 | 11,793 | 20,882 | 18,295 | 10,448 |
| Individual missing data <25% | 73 | 4 | 4 | 4 | 4 | 62 | 4 | 3 | 3 | 4 |
| Minor allele frequency >5% | 11,448 | 3,431 | 4,440 | 3,769 | 3,338 | 48,355 | 11,793 | 20,881 | 18,295 | 10,448 |
| Loci missing data <10% | 10,215 | 3,345 | 4,334 | 3,688 | 3,225 | 45,367 | 10,329 | 20,880 | 18,295 | 8,485 |
| On-target | 8,082 | 1,445 | 2,872 | 2,384 | 1,465 | 38,499 | 5,939 | 15,597 | 13,560 | 5,338 |
| Overall efficiency (%) | 102.15 | 33.45 | 43.34 | 36.88 | 32.25 | 90.73 | 20.66 | 41.76 | 36.59 | 16.97 |
| On-target efficiency (%) | 80.82 | 14.45 | 28.72 | 23.84 | 14.65 | 77.00 | 11.88 | 31.19 | 27.12 | 10.68 |

**Table 5 Average identity by state (±standard deviation) between replicates within and between the 10 and 50k assays for all species/subspecies of mountain sheep genotyped.**

| Species/ subspecies | Within 10,000 SNP assay | Within 50,000 SNP assay | Between 10,000 and 50,000 SNP assays | | | | |
|---|---|---|---|---|---|---|---|
| | RM bighorn | RM bighorn | RM bighorn | Dall | Desert bighorn | SN bighorn | Stone |
| IBS | 97.02 ± 0.03 | 95.99 ± 0.02 | 96.51 ± 0.04 | 94.84 ± 0.01 | 95.50 ± 0.02 | 95.06 ± 0.01 | 92.69 ± 0.01 |

**Table 6 Average identity by state (±standard deviation) within the 10,000, within the 50,000, and between the 10,000 and 50,000 SNP assays and their respective genotyping successes for differing sample types.**

| Sample type | Within 10,000 SNP assay | | Within 50,000 SNP assay | | Between 10,000 and 50,000 SNP assays | |
|---|---|---|---|---|---|---|
| | IBS (%) | Samples lost (%) | IBS (%) | Samples lost (%) | IBS (%) | Samples lost (%) |
| Blood | 98.94 ± 0.01 | 0 | 96.89 ± 0.01 | 0 | 97.88 ± 0.01 | 0 |
| Faeces | 93.48 ± 0.03 | 21.19 | 91.68 ± 0.02 | 56.25 | 88.55 ± 0.19 | 56.25 |
| Tissue | 98.58 ± 0.01 | 0 | 96.62 ± 0.01 | 0 | 97.56 ± 0.01 | 0 |

and 84% of genetic distance between populations (10k assay $r^2 = 0.88$, $p = 0.008$; 50k assay $r^2 = 0.84$, $p = 0.017$), respectively. Population structure was also reflected in the distribution of samples in PCA plots (Fig. 7). Notably, the two points positioned between Cadomin Mountain and Ram Mountain are genotypes from the same individual from Ram

**Table 7 Observed heterozygosity, expected heterozygosity, and Wright's inbreeding coefficient (*f*) for each sampling location of Rocky Mountain bighorn sheep.**

| Sampling location | 10,000 SNP assay | | | | 50,000 SNP assay | | | |
| --- | --- | --- | --- | --- | --- | --- | --- | --- |
| | Number of genotypes | $H_{obs}$ | $H_{exp}$ | $f$ | Number of genotypes | $H_{obs}$ | $H_{exp}$ | $f$ |
| Cadomin Mountain | 12 | 0.283 | 0.333 | 0.151 | 6 | 0.272 | 0.334 | 0.185 |
| Castle Yarrow | 16 | 0.301 | 0.332 | 0.094 | 16 | 0.292 | 0.333 | 0.124 |
| Narraway | 16 | 0.288 | 0.332 | 0.132 | 16 | 0.278 | 0.333 | 0.165 |
| Ram Mountain | 16 | 0.325 | 0.332 | 0.019 | 16 | 0.334 | 0.333 | −0.004 |
| Stornoway Mountain | 13 | 0.240 | 0.333 | 0.279 | 8 | 0.228 | 0.334 | 0.317 |
| Overall | 73 | 0.290 | 0.332 | 0.128 | 62 | 0.289 | 0.333 | 0.132 |

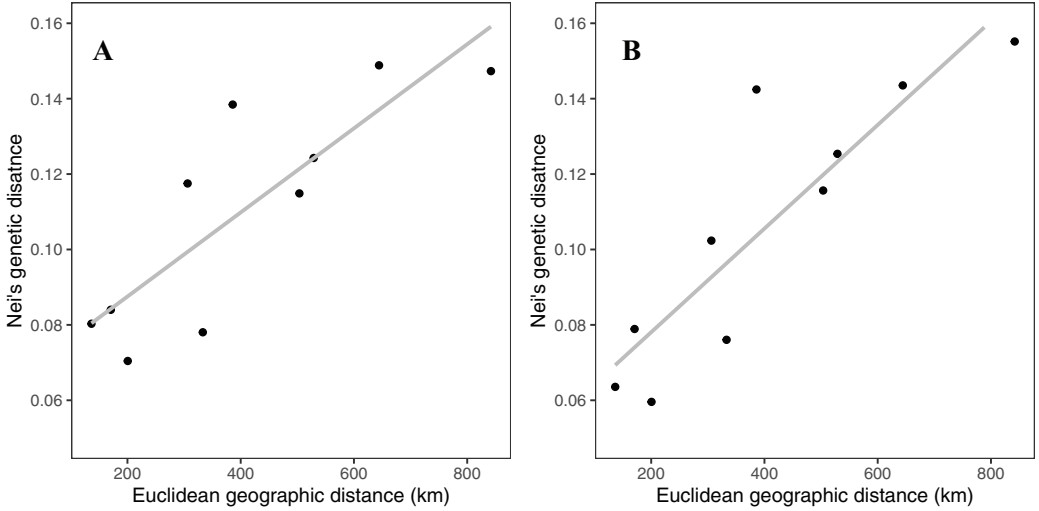

**Figure 6 Patterns of isolation-by-distance for Nei's standard distance and Euclidean geographic distance between the five Rocky Mountain bighorn sheep sampling locations.** Genotypes from the 10k SNP assay (A) and 50k SNP assay (B).

Mountain that had Cadomin-admixed ancestry according to pedigree data (*Coltman et al., 2002*; *Poirier et al., 2019*).Furthermore, as we expected points from replicated genotypes appeared as overlapping points or in extremely close proximity.

## DISCUSSION

We set out to develop and test the efficiency and accuracy of a species-specific high-density SNP genotyping assay for Rocky Mountain bighorn sheep. The final assay targeted 50,000 loci, which were evenly selected from chromosomes relative to chromosome size and were distributed throughout the genome with an average inter-SNP distance of 51,663 ± 60,107 bp. The 50k assay used with the ATG technology yielded overall and on-target efficiencies of 91% and 77%, respectively, with an average accuracy between replicates of 97%. Furthermore, our 50k assay had high concordance with previously generated genotypes (*Miller, Hogg & Coltman, 2013*; *Miller, Festa-Bianchet & Coltman, 2018*) with an average accuracy of 96.5% between these studies and our 50k assay. As we expected, we found

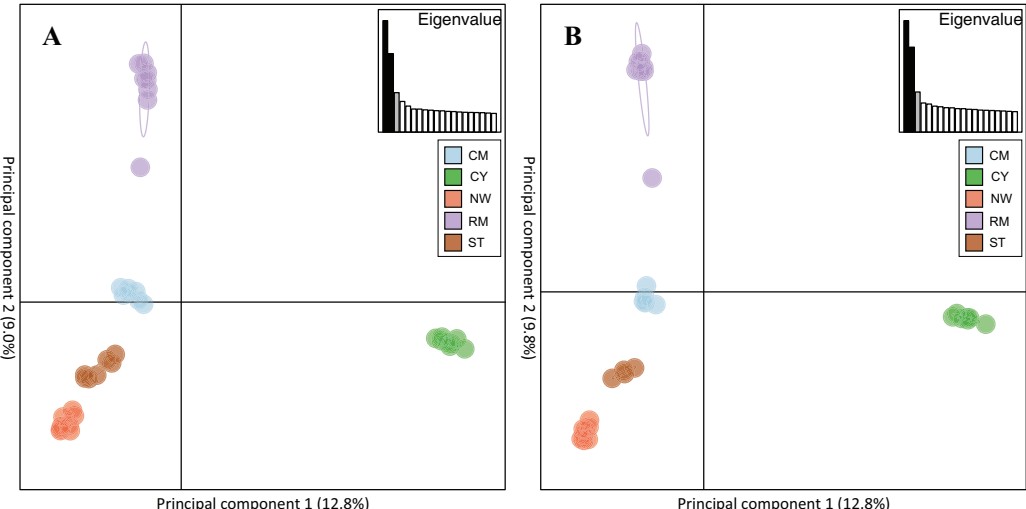

**Figure 7 Principal component analysis of the five Rocky Mountain bighorn sheep sampling locations.** Cadomin Mountain (CM), Castle Yarrow (CY), Narraway (NW), Ram Mountain (RM), Stornoway (ST), genotyped by the 10k SNP assay (A) and 50k SNP assay (B). As noted in the methods and results individuals were genotyped in duplicate, hence each individual successfully genotyped in duplicate will show two points on the PCA.

that genotype accuracies increased with better quality biological material for DNA extraction. Therefore, genotypes for DNA extracted from blood and tissue samples had a higher accuracy than those generated from faecal samples. Additionally, of the three sample types, only faeces derived genotypes were missing sufficient data for samples to be excluded from our final genotype dataset. Overall, the assay and technology yielded ample SNP data with a high degree of accuracy, with varying levels of accuracy and genotyping success from both tissue samples, but with lower genotyping success for faecal derived samples. However, we must note this SNP panel was developed using resources from a relatively small number of genomic studies from a few populations of bighorn sheep and was only trialed in Western Canadian populations, hence there may be some ascertainment or validation bias in our SNP panel and resulting genotypes.

## Faecal genotyping

Both assays exhibited lower success using DNA extracted from faecal samples than DNA extracted from other sample types. The 10 and 50k assays only retained ~79% and ~43% of faecal samples, respectively, and had accuracies of 93% and 92%, respectively. These values are low relative to the overall genotyping success of 85% and accuracy of 99.9% reported by *Gavriliuc et al. (2022)* for ATG genotyping faecal samples from domestic horses (*Equus caballus*). However, there are methodological differences between the studies. First, *Gavriliuc et al. (2022)* used swabs from fresh faecal samples, whereas our faecal samples were in the environment for an undetermined amount of time prior to collection. Secondly, *Gavriliuc et al. (2022)* typed 48 samples at 279 SNPs using a sequencing kit which provided ~800,000 reads, and thus had a sequencing coverage almost three times

greater than the 50k assay. With greater sequencing coverage we may have observed higher success and accuracy.

Unlike *Gavriliuc et al. (2022)*, our samples were exposed to the environment for an undetermined amount of time. During this time samples would have been exposed to ultraviolet light and, despite most of our samples being collected in the winter, potentially above optimal temperatures. Thus, our samples may have experienced more environmental degradation than those of *Gavriliuc et al. (2022)*. We also previously found a high rate of microsatellite genotyping failures using the same samples (*Deakin et al., 2020*), necessitating typing each sample in triplicate to generate reliable results. However, despite these issues, sampling fresher material is not easily done, given that bighorn sheep inhabit difficult to access terrain in remote locations.

There are two options for improving genotyping success and accuracy. The first is to increase sequencing depth. Samples in the 10 and 50k assay were sequenced with 125X and 20.8X, respectively, and faecal samples in the 10k assay had elevated accuracy (+1.8%) and genotyping success (+26%) over the 50k assay. This trend is also reflected in the accuracies and genotyping success for the two other sample types (Table 6) and relative to *Gavriliuc et al. (2022)*. This suggests increasing sequencing coverage also increases genotyping rate and accuracy in faecal samples, thus we recommend sequencing samples prepared by the ATG protocol with as much depth as feasible. Secondly, a challenge may be the presence of polymerase chain reaction (PCR) inhibitors in DNA samples extracted from faecal samples. PCR inhibitors may not be removed in DNA purification (*Morin et al., 2001*) and thus may inhibit efficient amplification. The ATG protocol used in this paper uses PCR, so it is possible PCR inhibitors have contributed to the poor performance of our assays on faecal material. Rectifying this by using different methodological approaches such as DNA extraction using magnetic beads (*Flagstad & Stacy, 1999*), or a post extraction clean-up such as ethanol precipitation (*Green & Sambrook, 2016*) may further also improve genotyping success and accuracy.

## Population structure

To test the utility of the assays for population genomic studies of Rocky Mountain bighorn sheep, we visualized the genetic similarity among individuals using Principal Component Analysis and by plotting isolation by distance. As expected, Rocky Mountain bighorn sheep samples grouped together by sampling location, and sampling locations separated relative to geographic separation. We also observed a strong pattern of isolation-by-distance among sampling locations using both assays, with over 80% of variance explained by geographic distance alone. This high degree of isolation-by-distance was expected, as previous analyses of bighorn sheep in this region identified a strong pattern of isolation-by-distance (*Deakin et al., 2020*).

## Application to other mountain sheep

Our assays had varying efficiencies with other mountain sheep species and subspecies. The variation in efficiencies appears to be explained by the phylogenetics of North American wild sheep. Efficiency decreases with time since divergence between the test and

target taxon. This was expected, *Miller, Hogg & Coltman (2013)* found that cross species applications of SNP assays yield exponentially less polymorphic SNPs as time to divergence increases. Of the sub-species of bighorn sheep, desert bighorn had the highest efficiency, which is expected given desert and Rocky Mountain bighorn sheep diverged more recently than Rocky Mountain and Sierra Nevada bighorn sheep (*Buchalski et al., 2016*). We see a further reduction in efficiencies for Dall and stone sheep, with efficiencies of ~33% and ~32% for the 10k SNP assay and ~21% and ~17% for the 50k SNP assay, respectively, which is expected given that bighorn and thinhorn sheep diverged ~1 mya (*Rezaei et al., 2010*). Despite reduced efficiency, genotype accuracy for all taxa were high, in the range of 93–96%.

## CONCLUSION

In conclusion, we successfully developed a high-density and accurate SNP assay for consistently genotyping Rocky Mountain bighorn sheep at ~45,000 SNP loci evenly distributed throughout their genome. The assay performs well on DNA extracted from tissue, but less so when used on DNA extracted from faeces. Thus, this assay enables us to perform high-throughput genotyping on Rocky Mountain bighorn sheep at a higher density than offered by previously used technologies (*Miller, Festa-Bianchet & Coltman, 2018*, *Miller et al., 2012*) and will serve as a genomic resource for future studies on the species. However, depending on the research goals, the lower density 10,000 SNP assay may be sufficient for examining population genetic structure or assessing individual relatedness and may be more reliable for use with degraded samples. Furthermore, this assay can be used to analyse many SNP loci in other species and sub-species of mountain sheep. The assay's efficiency decreased as the divergence times of these species and subspecies from Rocky Mountain bighorn sheep increased. However, the number of loci in the assay still allows for tens of thousands of SNP loci to be analysed. Thus, potentially having genetic and genomic applications in other North American species and sub-species of mountain sheep.

The development of this assay was facilitated by the cross-species application of the domestic sheep genome (*Bunch et al., 2006*) to Rocky Mountain bighorn sheep, further exemplifying the utility of genomic resources for studies of closely related wild species (*Li et al., 2019*; *Santos et al., 2021*; *Sim & Coltman, 2019*). Additionally, our study contributes to the growing number of successful applications of ATG and SPET in non-human mammals (*Andrews et al., 2021*; *Gavriliuc et al., 2022*) and other species (*Saber et al., 2017*; *Scaglione et al., 2019*; *Vu et al., 2023*), proving to be a viable option for low-cost, high-throughput, high-density SNP genotyping. Furthermore, targeted-GBS technology (*Kozarewa et al., 2015*; *Meek & Larson, 2019*; *Scaglione et al., 2019*) allows researchers to target specific loci throughout a genome enabling researchers to obtain relatively evenly distributed genome wide SNP markers required for quantitative genetic and genomic studies.

### Funding

This research was funded by a Natural Sciences and Engineering Research Council of Canada discovery grants to David W. Coltman and the Canadian Mountain Network. The funders had no role in study design, data collection and analysis, decision to publish, or preparation of the manuscript.

### Grant Disclosures

The following grant information was disclosed by the authors:
Natural Sciences and Engineering Research Council of Canada.
Canadian Mountain Network.

### Competing Interests

The authors declare that they have no competing interests.

### Author Contributions

- Samuel Deakin conceived and designed the experiments, performed the experiments, analyzed the data, prepared figures and/or tables, authored or reviewed drafts of the article, and approved the final draft.
- David W. Coltman conceived and designed the experiments, authored or reviewed drafts of the article, and approved the final draft.

### Animal Ethics

The following information was supplied relating to ethical approvals (*i.e.*, approving body and any reference numbers):

University of Alberta, University of Calgary, and Université de Sherbrooke

### Data Availability

The 50,000 SNP loci targeted by our SNP assay is available in the Supplemental Files.

The raw sequencing read data for the 10K and 50K SNP assay sequencing runs and genotypes from both the 10K and 50K SNP assays are available at the Federated Research Data Repository: Deakin, S., Coltman, D. (2024). Genotype and raw sequencing data for "Development of a high-density sub-species-specific targeted SNP assay for Rocky Mountain bighorn sheep (*Ovis canadensis* canadensis)". Federated Research Data Repository. https://doi.org/10.20383/103.0859.

### Supplemental Information

Supplemental information for this article can be found online at http://dx.doi.org/10.7717/peerj.16946#supplemental-information.

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
