# Peer review of "Development of a high-density sub-species-specific targeted SNP assay for Rocky Mountain bighorn sheep (Ovis canadensis canadensis)"

_PeerJ, doi:10.7717/peerj.16946_

## Round 0.1 · original submission · Major Revisions

Thanks for submitting your work to PeerJ.
Please make changes following the reviewers' comments.
Regards
LIN

Reviewer 1 ·

Basic reporting

My specific comments by line are below:
Line 24: It does not make sense as currently written that SNPs are "less informative" than microsatellites. Please clarify.
Line 337-338: This is an incomplete sentence.

Experimental design

The study's stated goal was to generate a SNP chip for bighorn sheep. While this is an interesting goal, the approach was not consistent with this. In addition, the authors used data that was convenient and available, rather than generating a dataset needed to develop a SNP chip that would be informative for different subspecies of bighorn sheep. In general, it seems that the authors sought to create a SNP chip relevant for their specific study area (without stating this directly) and sought to show that this could incidentally be usable in other populations and subspecies. Given ascertainment bias of SNP chips, it is unclear what these other applications would mean and how others could benefit. These core concerns in the study's goals and interpretation of the results need to be addressed. My specific suggestions by line are below:

Line 54: What is the overall goal of the SNP chip? I.e. to derive genetic distance between certain populations or relatedness between individuals? This would set up the methods and interpretation of the results. This is currently unclear.

Line 65: The stated goal to develop a SNP chip for Rocky Mountain bighorn sheep is imprecise and inconsistent with the paper. They seem to be developing the chip for an unspecified target population. However, later in the paper they apply the chip to other subspecies beyond Rocky Mountain bighorn sheep. Please clarify.

Lines 74-77: Please explain how using many different types of data for different populations would be expected to affect the results.

Lines 99-102: "Target populations" are referenced here but not referenced or specified earlier in the paper. What are your target populations?

Lines 117-119: Why were these other subspecies not included in development if SNP chip performance for these was of interest?

Lines 183-187: Inbreeding coefficients were calculated, but there is no "true value" to compare them to, so how does this help with SNP chip development?

Lines 239-241: Based on this sentence, are individuals are repeated in the PCA plot? This seems confusing. and the reasoning behind this should be explained. If this is the case, this also needs to be explained in PCA figure description.

Validity of the findings

The SNP chip might be relevant in the populations included in its development, but given the limited amount of data used for this, the interpretation of how these SNPs could be useful in other areas is less clear. My specific comments by line are below:

Line 244-245: This sentence is misleading- it is not a species-wide SNP chip based on its development. It seems to have been developed for only one area and its overall relevance is unclear.

Line 267: How would you achieve greater sequencing coverage?

Line 290: Forensics were not mentioned as a goal in the paper. This information first appears in the discussion. Please provide further background, methods, and results to support the appearance of this topic.

Line 308-309: This seems to be an indirect outcome, rather than a stated goal from the onset that is a result of careful methodology. Please clarify how this study sought to achieve this from beginning.

Reviewer 2 ·

Basic reporting

This manuscript describes the development of a SNP panel to genotype Rocky Mountain bighorn sheep samples of varying quality. I believe this manuscript is well written, clear and the analyses are appropriate and sound. I have very few remarks and suggestions, that I list below. In summary, I believe that the authors could harness a bit further their data to understand if an even more reduced dataset would be appropriate for fecal samples, and use population genetic analyses to test for the power of such reduced dataset.
Congratulations on the very interesting work.
Main comments:
L258 – have you considered further reducing the SNP set for a few thousand loci and see if the genetic diversity indices and patterns would hold? In that case, it would be more likely to have increased proportion of genotyped loci in fecal samples. 10k is still a very high number of SNPs. It would be interesting to see something like a structure plot including the fecal samples at a reduced SNP set, just to have an idea of how well this dataset could perform even at lower number of SNPs. I understand that, if you have many fecal samples, the randomness of SNP amplification will likely not allow for such analyses, because increasingly fewer loci will be commonly genotyped the more fecal samples you include, and hence my suggestion of considering a smaller genotyping set, if you believe that fecal sampling is of great importance in this system.
I was wondering why didn’t you try to include samples from western BC, Washington or southern Idaho to test your SNP set, before considering other subspecies or species? It would have been interesting to see if this SNP set supported or not the distinction between Rocky Mountain and California bighorn sheep.
L290 – from the introduction, it is not very clear why forensics is mentioned here. Adding a few words to the intro to set up the need for forensic analysis would be important.
L307 - I think it is important to add a map with the sampling localities for readers unfamiliar with the geography of this region, also in order to understand the isolation by distance pattern

Small comments:
L173-174 – not sure what “at in common” means.
L221-223 – I would replace the semicolon by colon, as I believe you are listing the three types of IBS classes

Experimental design

no comment

Validity of the findings

no comment

Reviewer 3 ·

Basic reporting

• The manuscript is very well written, straightforward, and easy to follow. Professional English is used throughout.
• I recommend adding additional citations throughout the introduction. For example, there are many possible manuscripts that could be cited to support the argument that “… single nucleotide polymorphisms (SNPs) are a common marker-type of choice for contemporary population genetic and genomic studies…” (page 6, line 22).
• Citations should probably be checked for formatting. For example: 1) on page 18, lines 374-375, scientific name should be italicized and 2) on page 22, line 476, author name should not be completely capitalized.
• The introduction is well-written and sufficient background/context is provided. I might recommend including information on relatedness amongst species/subspecies in the last paragraph of introduction, rather than saving this information entirely for the discussion. Could a phylogenetic tree be added as a figure?
• The article structure and figures are professional. In general, I think it would be helpful if the authors provided more detail in the figure legends. For example, in the legend associated with Figure 4, authors might specify how inter-locus distances are calculated. I tried to find this is the methods section but could not.
• The tables are professional… but tables 4-7 are also labeled Table 1? (directly above each table)
• I could not find all relevant raw data. There is an Excel file describing the loci genotyped (a legend or README file would be helpful), but the genotypes associated with each sample are not included and neither are the raw sequencing files. I could not find a data availability section in the body of the manuscript.
• The manuscript is self-contained and the results are relevant to the hypotheses.

Experimental design

• This research article describes original primary research within the Aims and Scope of the journal. The research question is clearly stated in the abstract and introduction (whether a model species genome can be used to design a SNP assay for a closely-related, non-model species). This manuscript will be useful to other scientists/conservation managers using high throughput sequencing and bioinformatic approaches to study non-model species.
• The investigation was performed to a high technical & ethical standard. The methods are described with sufficient detail and information to replicate. I did wonder whether the material on page 13, lines 225-231 should be excluded from the manuscript. The authors ultimately used these data to argue that “…our 50K assay had high concordance with previously generated genotypes (Miller et al. 2013, Miller et al. 2018) with an average accuracy of 96.5% between these studies and our 50K assay.” But, only two samples from the previously studies were included (lines 225-231) and there are a number of methodological issues that could influence the “96.5% accuracy”. I think these data/arguments are a bit shaky, and the manuscript is stronger without them.

Validity of the findings

• Very helpful, meaningful replication (amongst sample types, methodological approaches, populations) is included in the experimental design.
• I could not find all relevant raw data. There is an Excel file describing the loci genotyped (a legend or README file would be helpful), but the genotypes associated with each sample are not included and neither are the raw sequencing files. Otherwise, the experimental design is robust and conclusions are statistically sound.
• Conclusions are well stated, linked to original research question & limited to supporting results. I have a couple suggestions that I believe would improve the manuscript. 1) I think it would be helpful to include the per-sample cost using this approach to SNP genotyping (or a relevant citation). This would be helpful for scientists considering adopting the approach. 2) As greater coverage is preferred and easier (or at least less expensive) to obtain with the 10K SNP approach, are there any scenarios in which you would recommend using the 50K SNP panel instead? It seems that 10K would be more than enough for evaluating population structure, generating pedigrees, testing for loci under selection, etc. and would increase the number of samples that would pass quality control measures. I think a discussion of this would be helpful.

---

## Round 0.2 · Minor Revisions

Please reply the Reviewer 1's comments, and explain why you did not positively reply to their comments.

Reviewer 1 ·

Basic reporting

Thank you for clarifying throughout the paper that the goal was to evaluate the sub-species of Rocky Mountain bighorn sheep- this was not clear in the initial work. I think it needs to be further clarified that the tool is likely to be most informative for those populations included in its development.

Experimental design

The authors still need to provide a justification for why the same animal is shown in duplicate in the PCA. Are the points for the same animal overlapping or are the results different for the same animal using the new assay? Perhaps they could provide a summary table showing how the same samples had consistent or different results, rather than plotting them as overlapping points in a PCA, which would be a more helpful demonstration of the assay's consistency.

Validity of the findings

The phrase "limited data" in my previous comments was a reference to the limited number of populations out of the entire subspecies that were included in your analysis, which was also pointed out by Reviewer 2. While you were able to include SNPs from the studies that had SNPs available for Rocky Mountain bighorn sheep, this is still a small subset from large number of populations of the subspecies. One would expect different markers to be variable in different populations. Thus it would be helpful to clarify that the assay is likely to be informative for those populations that were included in the assay's development.

Additional comments

I would like to see the authors address the original questions from the reviewers more thoroughly. For example, I agree with Reviewer 3 that the approximate cost of the assay would be a helpful addition. Most questions were intended to be helpful for clarification for the reader, such as clarifying their study goals. They often chose to respond bluntly in the rebuttal, rather than enhance their paper with the help of this feedback.

Reviewer 2 ·

Basic reporting

I believe the authors answered my previous questions in a satisfactory manner. I still beleive there is ano particular issue that I would like to see better layed out, and which I detail in the following sections.

Good job!

Experimental design

I would still caution the authors regarding the fact that they developed an assay based on only a few populations of the entire remaining 'native' Rocky Mountain populations available. Thus, there should be an ascertainment bias toward Edmonton samples, considering that the entire distribution of Rocky Mountain (including populations that were never extirpated) extends further south. This is especially important given the history of accentuated decline in this species, with multiple instances of sharp bottlenecks, drift, founder effects, and also admixture (which could diminish this bias).

Validity of the findings

Although I do not think this should require the authors to change the title, I do believe ascertainment bias should be mentioned in the manuscript, so that other researchers are aware of samples to include in the future to ensure their populations are equally well represented. I suggest specifically including a map of all Rocky Mountain native populations with a highlight on those sampled for this analysis.

Reviewer 3 ·

Basic reporting

This manuscript meets all the requirements associated with basic reporting.

Experimental design

This manuscript meets all the requirements associated with experimental design.

Validity of the findings

This manuscript meets all the requirements associated with validity of the findings.

Additional comments

The authors have largely addressed my concerns. I do not think the manuscript needs additional peer review prior to publication.

However, I disagree with the argument "Indeed, the per sample cost is important. However, the technology used for the approach is provided by a company which regularly changes their prices/offers different pricing schemes, hence our per sample cost would not be relevant in a years (maybe less) time." New sequencing technologies arise rapidly, so given the authors' logic, it would not make sense to publish this manuscript at all, as a new approach may make SPET assays obsolete within a year.

I would very much have appreciated it if the authors had simply included the per-sample cost.

---

## Round 0.3 · accepted · Accept

Congratulations!

Your work is better. Thanks for submitting your work to PeerJ.